## PERSPECTIVE

# The oncological relevance of fragile sites in cancer

Benjamin S. Simpson [1], Hayley Pye [1] & Hayley C. Whitaker [1✉]

Recent developments in sequencing the cancer genome have provided the first in-depth mapping of structural variants (SV) across 38 tumour types. Sixteen signatures of structural variants have been proposed which broadly characterise the variation seen across cancer types. One signature shows increased duplications and deletions at fragile sites, with little association with the typical DNA repair defects. We discuss how, for many of these fragile sites, the clinical impacts are yet to be explored. One example is *NAALADL2*, one of the most frequently altered fragile sites in the cancer genome. The copy-number variations (CNVs) which occur at fragile sites, such as *NAALADL2*, may span many genes without typical DNA repair defects and could have a large impact on cell signalling.

Fragile sites are specific loci that are vulnerable to breaks and constrictions when chromosomes are exposed to replication stress, acting as genomic 'fault lines'[1]. Recently, Li et al. provided the most detailed account of structural variants (SV) in the cancer genome to date where researchers derived 16 distinct signatures of structural rearrangement. The signatures were characterised by an over-representation of a particular SV class, size, replication timing and genomic location. They also compared the co-occurrence of these signatures with known pathogenic mutations in key DNA repair genes (e.g. *ATM, BRCA1, BRCA2*). The fragile sites signature showed only moderate co-occurrence with alterations in DNA repair genes, instead being characterised by deletions and tandem duplications at chromosomal fragile sites[2]. The genes in closest proximity to the most commonly affected fragile sites highlighted by Li et al. are shown in Table 1[2]. While the mechanism underpinning these sites is not fully understood, there are a number of proposed mechanisms for why these fragile sites are so vulnerable to breaks.

**Fragile sites associate with extreme genomic stress**. Large genes are considered more likely to harbour fragile sites[3], with the most common fragile sites (CFS) corresponding to the largest actively transcribed genes or transcription units (TU) in both human and mouse cells[4]. Active TUs >1 Mb are known to be reliable predictors of chemically induced CNV hotspots[4]. In the most affected fragile sites described by Li et al., all genes were ≥0.89 Mb with an average length of 1.54 Mb (Table 1)[2]. The transcription-dependent double-fork failure (TrDoFF) model proposes that genomic instability may arise from cellular stress induced by transcription during replication[5–7]. Curiously, the increased transcription in large TUs does not necessarily increase the instability and may even increase the stability at these sites[4,8]. The TrDoFF model suggests that large TUs could promote simultaneous failure of two converging replication forks through the formation of RNA:DNA hybrid structures known as R loops[5]. Alternatively, large TUs may create late-replicating domains, which prolong transcription into the S-phase and disrupt the initiation of DNA replication at origins (origin firing)[5]. As large TUs fail to replicate the DNA in the S-phase, these regions have also been shown to uniquely exhibit mitotic DNA synthesis (MiDAS)[9,10]. Sites of MiDAS may be defined through a method known as MIDAS-seq and are evident as well-defined twin peaks that merge into a single peak as the M-phase progresses[11]. These peaks are conserved between cell lines and encompass all known CFSs as well as regions resembling CFSs[11]. Consequently, the presence of MiDAS is an indicator that cells are experiencing DNA replication stress[9]. Within these unreplicated regions, fragile site breaks occur,

[1] Molecular Diagnostics and Therapeutics Group, Research Department of Targeted Intervention, Division of Surgery & Interventional Science, University College London, London, UK. ✉email: Hayley.whitaker@ucl.ac.uk

**Table 1 Most commonly altered fragile sites ranked from most to least affected by structural variation in the cancer genome.**

| Gene containing fragile sites | Genomic co-ordinates | Size (Mb) |
| --- | --- | --- |
| FHIT | Chr3:59747277-61251459 | 1.50 |
| MACROD2 | Chr20:13995369-16053197 | 2.06 |
| WWOX | Chr16:78099430-79212667 | 1.11 |
| IMMP2L | Chr7:110662644-111562517 | 0.90 |
| NAALADL2 | Chr3:174438573-175810548 | 1.37 |
| LRP1B | Chr2:140231423-142131016 | 1.90 |
| PDE4D | Chr5:58969038-60522120 | 1.55 |
| CCSER1 | Chr4:90127535-91601913 | 1.47 |
| DMD | ChrX:31097677-33339441 | 2.24 |
| PACRG; PARK2 | Chr6:161740845-163315492 | 1.57 |
| KIF26B; SMYD3 | Chr1:245353678-246507312 | 1.15 |
| PTPRD | Chr9:8314246-10612723 | 2.30 |
| LSAMP | Chr3:115802363-117139389 | 1.34 |
| AUTS2 | Chr7:69598296-70793506 | 1.20 |
| RBFOX1 | Chr16:5239802-7713340 | 2.47 |
| CSMD1 | Chr8:2935353-4994972 | 2.06 |
| PRKG1 | Chr10:50990888-52298423 | 1.31 |
| DIAPH2 | ChrX:96684663-97604997 | 0.92 |
| NEGR1 | Chr1:71395943-72282539 | 0.89 |
| GPC6 | Chr13:93226807-94408020 | 1.18 |
| CTNNA3 | Chr10:65912523-67696195 | 1.78 |

This list was transcribed from the Li et al. extended data Fig. 9B2. The semicolon indicates a fragile site between two adjacent genes. Genomic co-ordinates are shown for each site (mapped to the GRCh38.p13 reference genome).

creating a deletion CNV in the DNA that spans the TU[5]. This is supported by experimental evidence in primary cells that shows clusters of double-stranded gene breaks and translocations that localise to the gene bodies of longer genes[12]. The alternate possibility is the formation of a copy-number gain.

Okazaki fragments are short sequences of DNA nucleotides (150–200 base pairs long in eukaryotes) that are synthesised discontinuously on the lagging strand. At fragile sites, duplications (CNV gains) may also occur, theoretically following fork-stalling, when the $3'$ end of a nascent Okazaki fragment disengages and anneals with the lagging strand template of a nearby replication fork undergoing replication[13]. This is known as the fork stalling and template switching (FoSTeS) model[14]. In more contemporary work, the FoSTeS model is superseded by the microhomology-mediated break-induced replication model, which proposes that a single double-strand end results from replication fork collapse in a cell under stress and as part of the stress response, repair molecules RecA/Rad51 become down-regulate preventing double-stranded repair. As a consequence, the $3'$ end from the collapsed fork anneals to any single-stranded template with sufficient microhomology. This annealing typically occurs in front of, or behind the position of the fork collapse, leading to gene deletion or duplication, respectively[14].

The alternative breakage–fusion–bridge cycle model proposes that double-strand breaks between the DNA are bridged, joining the Watson and Crick strands, and that, over progressive cycles of breakage and fusion, create a series of tandem inverted gene duplications[13]. However, the exact mechanism behind these gene duplications remains unclear and a number of other plausible models exist[13].

Irrespective of the mechanism, experimental evidence shows a clear correlation between fragile sites and copy-number changes[4,14]. In cell models, genome instability occurs in cells treated with DNA replication-stress-inducing agents, eventually resulting in CNVs in the genome[15]. Mapping of the resulting CNVs follows these genomic fault lines and large genes, including those identified in the Li et al. study[2,15]. While deletions at these loci are more common with chemically induced replication stress, gains have also been observed in cells[15]. If these alterations provide a fitness advantage, then it seems feasible that the frequency of alterations may increase through clonal selection.

Many of the genes that harbour these fragile sites and CNVs have already been implicated in oncogenesis and have well-established roles in cancer development and/or progression, e.g. the tumour suppressors FHIT and WWOX[16]. However, some sites are poorly understood, such as the site at the N-acetylated alpha-linked acidic dipeptidase like-2 (NAALADL2) gene.

**The fragile site in NAALADL2 may have a functional role in tumourigenesis.** NAALADL2 was identified as the fifth most altered site in a pan-cancer analysis by Li et al. It is a giant gene spanning 1.37 Mb, approximately 45 times larger than the average gene, which is usually between 10–15 kbp[17,18]. The biological role of NAALADL2 and its relevance in oncogenesis are relatively understudied. However, data exist implicating NAALADL2 in tumour development and progression[19–23].

Genome-wide association studies (GWAS) have linked single-nucleotide polymorphisms (SNPs) in NAALADL2 to risk in breast and lung cancers and several studies have identified SNPs within the NAALADL2 locus that are associated with prostate cancer risk or aggression[20,22–25]. A GWAS of 12,518 prostate cancer cases identified rs78943174 within the NAALADL2 locus as one of two loci associated with a high Gleason sum score,[22] leading to suggestions that NAALADL2 could be a potentially valuable therapeutic target[21]. Other SNPs in NAALADL2 have been found in TP53 and GATA2 binding sites and associated with reduced time to biochemical recurrence in patients undergoing radical prostatectomy[20,25]. SNPs within the NAALADL2 locus have been shown to be in linkage disequilibrium (LD) with SNPs associated with an increased risk of PCa, suggesting possible synergy or, alternatively, that one of these genes represents a false-positive association[26].

NAALADL2 protein expression has previously been shown to be increased in higher-stage and grade cancers[27]. Its overexpression in prostate cancer cell lines can lead to altered extracellular matrix binding, increased growth and invasive capabilities. Cell lines overexpressing NAALADL2 had altered

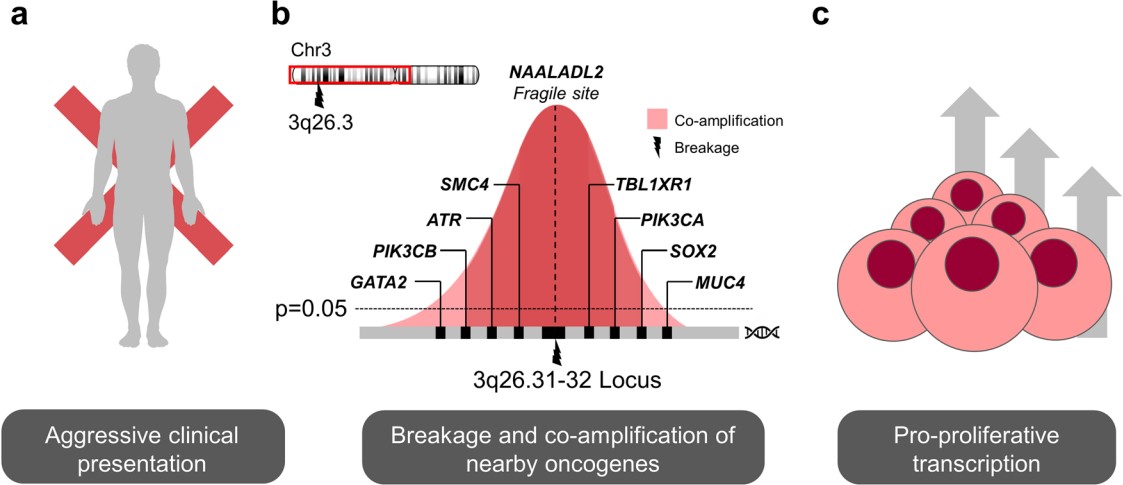

**Fig. 1 Overview of evidence surrounding the fragile site *NAALADL2*'s association with aggressive PCa. a** The Frequency of *NAALADL2* amplifications increases with Gleason grade, tumour stage and local metastasis in PCa. **b** Upper: Location of *NAALADL2* on Chromosome 3; The lightning symbol indicates the location of the fragile site. The red box indicates the extent of the region that can co-amplify with the *NAALADL2* genomic region surrounding 3q26.31, which is rich in oncogenes. Lower: pictograph displaying nearby oncogenes co-amplified with *NAALADL2* in PCa. The x-axis shows the genomic location of genes within the amplicon, the y-axis represents significant co-occurrence (−Log10 *p*-value). **c** Increased copy number results in increased transcription of oncogenes through the 'gene dosage' effect as well as downstream activation of other oncogenes. The diagram shows tumour cells replicating following a number of pro-proliferative mRNA signalling pathways becoming activated.

transcription of genes in pathways involving the cell cycle, cell adhesion, epithelial to mesenchymal transition and cytoskeletal remodelling, suggesting a potential functional role in tumour progression; however, the specific nature of its mechanism remains elusive[27].

We recently published a report on the association of somatic copy-number gains at the *NAALADL2* locus with an aggressive prostate cancer phenotype[28]. Copy-number gains in *NAALADL2* were found to occur in 15.99% (95% CI:13.02-18.95) of primary prostate cancers with increasing frequency in metastatic, castrate-resistant and neuroendocrine disease. This contrasts the pattern of *NAALADL2* CNVs across all tumour types, where the loss occurred more frequently than gains[2]. Gains in *NAALADL2* were associated with clinical hallmarks of aggressive prostate cancer, including tumour stage, Gleason grade, reduced time to disease recurrence following radical prostatectomy, increased likelihood of a multi-focal tumour, positive surgical margins and lymph node metastasis[28]. Importantly, of the 465 genes that were frequently co-amplified with this locus, 47.5% of the genes displayed a significant increase at a transcriptional level compared to just 2.36% that were downregulated[28]. This suggests that a gain or loss may have a predictable effect on transcription and therefore the function of any affected gene is important. Copy-number gains in the locus co-occurred with 67 nearby oncogenes, including *BCL6*, *ATR*, *TERC* and PI3K family members, and are associated with the altered transcription of 473 oncogenes, activating pro-proliferative transcription processes[28]. Therefore, the consequences of potential breakage at these fragile sites can be highly significant.

Ren et al. proposed a small signature of five proteins (Ki-67, Cyclin E, POLD3, γH2AX and FANCD2) associated with DNA replication stress across several tumour types[29]. We observed significant (albeit small) increases in the corresponding mRNA transcripts of the genes encoding these proteins: MKI67 (Log2 FC: 0.63, *paj* = 0.000059), *CCNE1* (Log2 FC: 0.29, *paj* = 0.018), *POLD3* (Log FC: 0.13, *paj* = 0.011) and *FANCD2* (Log2 FC: 0.35, paj = 0.000087) in patients with *NAALADL2* gain compared to diploid carriers (no changes in *H2AFX* expression)[28]. This supports the hypothesis that those patients with gains in this

region have increased replication stress. In the case of *NAALADL2*, this correlates with a CNV in a potentially clinically significant fragile site as summarised in Fig. 1 Given the large size of the *NAALADL2* gene, replication stress at this site may increase the chance of breakage and the formation of an SV. Alternatively, it may be that once a duplication event has occurred, transcription of such large transcripts could be responsible for increasing the replication stress.

Importantly, unlike *FHIT* and *WWOX*, it currently remains unclear whether the associations between the *NAALADL2* fragile site and this gene signature are related to the protein function of *NAALADL2*. This seems plausible given that as the locus surrounding the *NAALADL2* fragile site is rich in oncogenes, upon breakage, gains frequently co-occur, leading to concurrent changes in expression in pro-proliferative genes that could drive clonal expansion[28,30]. This raises the possibility that the location of a fragile site and the proximity of any oncogenes may be used to predict its significance in disease. The majority of research into the *NAALADL2* fragile site has been in prostate cancer. Furthermore, just as fragile site–CNA interaction is often cell-type specific, it is likely that fragile site SV signatures are specific to certain tumour types, and this could prove to be a worthwhile area of research. This is supported by the findings of Li et al., who noted that tumours of the gastrointestinal tract such as colorectal and oesophageal adenocarcinomas showed higher rates of the fragile site signature and overall prostate cancer showed little enrichment for the fragile site signature. It may prove useful to further assess these sites in theses specific tumour types[2].

## Conclusion

Tumours with copy-number changes that occur predominantly at fragile sites represent a distinct class of structural variation in the cancer genome. The clinical significance of many of these sites remains unexplored, as evidenced by the frequently altered fragile site within *NAALADL2* that has only recently attracted scientific interest. Research into this gene has highlighted the possibility that the function of the encoded protein may not be the only factor influencing the impact of structural variants. Given the

broader effects and scale of the CNVs that may occur along these fault lines in the absence of significant DNA repair defects, fragile sites are likely to represent important sites in the cancer genome that have so far been largely overlooked.

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

## Acknowledgements

The authors would like to acknowledge the support of University College London. Thanks to the John Black Charitable Foundation, The Urology Foundation and Rosetrees Trust for funding BSS. Additionally, we are grateful to Prostate Cancer UK and Movember for their support via the Prostate Cancer Centre of Excellence and TLD-PF16-004 grant that supports the work in our laboratory. Finally, we are very grateful to the patients who gave their samples for this research

## Author contributions

BSS: writing and development of the manuscript and figure design. BSS, HP, HCW: development of hypothesis and interpretation of literature. HP and HCW: proof-reading and editing of the manuscript. HCW: provision of funding.

## Competing interests

The authors declare no competing interests.
