## [Peer Review File · Communications Biology]

Reviewers' comments:

Reviewer #1 (Remarks to the Author):

The manuscript by Simpson et al. puts forth a hypothesis that replication stress can cause the formation of fragile sites (FS) thereby increasing copy number of genes, including oncogenes, within and near the FS, which can in turn provide an oncogenic fitness advantage to cancer cells. The authors first presented existing models for the role of common FS then argued for the importance of NAALADL2, a gene within a FS, in oncogenesis.

While I agree structural variants (SV) at common FS is an interesting and relatively under-studied class of somatic SVs, it is not entirely novel. There is a large body of work by Thomas Glover, Thomas Wilson, et al. on not only the relationship between common FS and copy-number aberrations (CNA) but also on the mechanism of this interaction on tumourigenesis.

Just as FS-CNA interaction is cell-type specific, it is likely that FS SV signatures are specific to certain tumour types, and this could prove to be a worthwhile area of research. Furthermore, because FS SV signatures are dominated by deletions (copy number loss) with well-described mechanistic models, the presence of copy number gains (tandem duplications) within FS is interesting, and possibly worthy of further investigation. However, while this paper provides a reasonable summary of existing work on FS and CNA in cancer genomics, and a call-to-arms for research into this area, it does not by itself add sufficient novelty to the field.

Two major claims were (re-)presented in this commentary, both of which were recently published by the same authors in the same journal (Simpson et al. 2020). Firstly, the authors reiterated that copy number gains in the FS gene NAALADL2 were found in 16% of localised prostate cancer (PCa) and is associated with aggressive disease. The mention of this claim is interesting in this commentary because it is not supported by the PCAWG (Li et al.) paper, which the authors heavily cited. In the PCAWG paper, CNA for NAALADL2 is similar to other FS genes in that SVs within FS are dominated by copy number losses rather than gains. Furthermore, no FS SV signature was observed in PCa in the PCAWG study (Sup Fig. 8). The second major claim is that copy number gains in the NAALADL2 locus co-occur with nearby oncogenes. Given linkage disequilibrium and the fact that genomic breaks at fragile sites could theoretically occur anywhere within a FS, it is unsurprising that there is co-occurrence of copy number gains of nearby genomic region. However, CNA does not always translate proportionately, or at all, to transcriptional changes. Thus, it is important to demonstrate whether there is **also** co-occurrence of increased gene/protein expression of nearby oncogenes.

Fundamentally, the authors are hypothesising that replication stress contributes to the formation of FS, particularly at NAALADL2, thereby increasing the copy number of nearby oncogenes, thus conferring oncogenesis. To demonstrate the validity of this hypothesis, the authors will need to (1) demonstrate replication stress contributes to FS, not the other way around, and (2) that copy number gain of common FS (e.g. NAALADL2) results not only in increased expression of NAALADL2 but also nearby oncogenes.

I am concerned that the proposed hypothesis that "replication stress may contribute to the formation of clinically significant FS" is overstated. Association is not causation. The data that's presented only support that copy number gain in NAALADL2 is **correlated** with increased protein levels that have been implicated in DNA replication stress. It does not imply that replication stress induces copy number gain of NAALADL2. From existing theories of other FS, it is equally plausible that CN gain of NAALADL2 may be the cause of replication stress. Similarly, the suggestion that gains at common FS can provide a fitness advantage, thereby cancer clonal selection, is in significant contrast to the alternate hypothesis that **loss** of common FS genes results in genomic instability causing CNA of cancer driver genes, which in turn confer selective advantage (e.g. see Saldivar et al. Plos Genetics, 2012; 8(11): e1003077). While it is important to challenge dogmas, it is important for the authors to recognise/acknowledge existing theory and to demonstrate whether it is complementary or mutually exclusive of the hypothesis they are proposing.

Findings from the authors previous studies are a good starting point but further experiments /

investigations are needed to support the current hypothesis.

If the authors are able to provide the suggested proofs, I would encourage a full article rather than a commentary.

Specific points:

- The authors stated that "The FS signature showed little co-occurrence with alterations in DNA repair genes...". This statement is not entirely correct as TP53 was shown to be mildly associated with 10 of the 12 most distinctive SV signatures in the PCAWG paper. Furthermore, it is known (e.g. Schepeler et al. *Oncogene*, 2013; 32: 3577-3586) that TP53 mutations can provide the necessary replicative stress and selective pressure for the outgrowth of clones with defective DNA damage-response.
- Table 1 is supposed to present commonly affected FS highlighted by the PCAWG (Li et al.) paper. However, the original paper reported 18 major fragile sites (in their supplementary information), while 21 are reported in this Table 1. Presumably that's because Table 1 was transcribed from Li et al's Extended Data Fig. 9B, but nonetheless, clarification on the discrepancy would be helpful.
- The authors indicated that "NAALADL2 was identified as the fourth most altered site in a pan-cancer analysis by Li et al., (2020)". However, Extended Data Fig 9 of Li's paper suggest NAALADL2 is the FS gene with the fifth most number of PCAWG samples (not fourth) and 6th to 8th most number of SVs (not fourth). Clarification on where the 4th ranking was derived would be helpful.
- The authors noted that "Given the large size of NAALADL2, an increase in transcription prior to replication through SNPs and/or altered transcription factor binding could theoretically increase replication stress and the chance of breakage within the gene due to the FS within it." However, wouldn't "gene breakage" imply a loss of function or copy number loss?

Reviewer #2 (Remarks to the Author):

Simpson et al. present a nicely written summary of the specific class of copy number alterations seen in cancer that arise at chromosome fragile sites (FSs). It is timely, as these genes continue to gain increasing attention and yet are difficult to fully understand regarding the impacts of alterations. These loci are hyper mutagenic and so mutations may or may not correlate with oncogenic function. In this context, the perspective offers a nice balance between an initial general consideration and a latter portion that dives more deeply into what I agree is a less well considered gene in this context, NAALADL2, but which is clear a site of hyper-mutation. Overall, the review is clear, interesting and accurate and I endorse it.

I do have a number of smaller comments below, but even in aggregate these are minor and offered for the authors' consideration.

1) Abstract: the phrase "may span hundreds of genes" is confusing as it is hard to reconcile a focal process (at a single gene such as NAALADL2) with an event that would cover large portions of a chromosome. For the claim to be compelling, I would expect it to be possible to know that such very large events were mechanistically or functionally linked to the one FS gene of interest, and I am not sure that is possible. Simply changing to "many genes" may ameliorate a potentially misleading phrase.

2) Table 1 and associated text. It is disappointing that the presentation uses the GRCh37/hg19 reference genome, instead of hg38. Transcript annotations associated with hg38 are significantly more accurate at many large genes, in fact leading us to understand that they are even larger than as tabulated (one example is LSAMP). hg38 has been the standard reference genome from some years now.

3) Page 2, line 52. I do not agree that the TrDoFF model referenced by the authors suggests that "genomic instability may arise from cellular stress induced by HIGH levels of transcription". In fact, Wilson et al. (ref 4) showed that the level of transcription did not correlate with instability in vitro,

a concept extended by Blin et al (not referenced, PMID: 30598553) who showed that high levels of transcription in fact reduced instability. The thought could be fixed by simply removing "high levels of".

4) Page 2, line 62. To be sure, I agree that the mechanisms of duplication formation are enigmatic, but I find the reference to template switching to the opposite side of the gene to be somewhat at odds with the most nuanced understanding of the FoSTeS and MMBIR mechanisms. Newer papers from Lupski and colleagues that proposed these mechanisms suggest that duplications arise when template switching occurs to a location BEHIND the stalled fork of interest. Since the forks usually stall within the genes, this predicts that most FS-associated duplications will occur on the flanks of those genes, which was observed in vitro by Wilson et al. For a switch behind a fork to lead to duplication of an entire FS gene, the fork would have had to have already passed through the entire gene, at which point replication would likely have already been resolved. Thus, duplications that include the entirety of a FS gene likely occurred by a mechanism other than instability of the FS gene itself (such as some of those listed). I'm not sure this requires too much text alteration, but perhaps the sense of how and where template switching is likely to occur at FS genes would help with clarity.

5) page 3 , line 92. "suggesting possible synergy". The fact that NAALADL2 genes are in disequilibrium with other genes associated with cancer might also support a hypothesis that one or the other are false positive associations, e.g. that the NAALADL2 alterations are not the drivers of an observed cancer association?

6) There are two points from the literature that are not referenced that I think are highly relevant to the discussion. I would recommend working them in if possible:

a) Wei et al (PMID: 26871630) used a DSB-driven assay to show that large genes are frequent sites of genomic alterations such as translocations and others. It stands as an excellent documentation of their increased instability in primary cells.

b) Macheret et al (PMID: 32561860) and others recently showed that large genes at FSs are uniquely prone to completing replication in M phase, known as Mitotic DNA Synthesis (MiDAS). I don't consider any discussion of FS instability any more without considering that highly exotic replication pattern.

Although overall the paper is clearly written, I point out a few places where I felt the grammatical usage could be improved:

a) page 1, line 31, "replication stress acting as genomic 'fault lines'". Sounds like replication stress is acting as a fault line (not the FS). Probably just put a comma after "stress".

b) page 3, line 86. "SNPs within the NAALADL2 locus associated with prostate cancer". Given the sentence structure, I might suggest adding "that are associated" to make it clear that it is the SNPs that are associated, not the gene.

c) page 3, line 82, "However, more frequently, supplementary data exists". I suggest deleting "more frequently, supplementary". It is not clear what is meant to "more frequent" or "supplementary" so those words seem superfluous. Also, "data" is a plural word, so it should be "exist", not "exists".

d) page 3 , line 90. "BCR" is never defined, I don't know what it is.

e) page 3 line 105. "NAALADL2 gains associated with". Add "were" before "gains". In general, you will see that I think the liberal use of the word "associated" was sometimes confusing without more words to guide the grammar.

Rebuttal for The fault lines in our genome: the oncological relevance of fragile sites in cancer

Reviewer 1:

While I agree structural variants (SV) at common FS is an interesting and relatively understudied class of somatic SVs, it is not entirely novel. There is a large body of work by Thomas Glover, Thomas Wilson, et al. on not only the relationship between common FS and copy-number aberrations (CNA) but also on the mechanism of this interaction on tumorigenesis.

We would like to thank the reviewer for their comments and taking the time to give constructive feedback. We would agree, the topics introduced in this article are not entirely novel (nor are they intended to be) and we gratefully acknowledge the work of others in this field. We have included more of this relevant literature (including that of Glover and Wilson *et al.*) in the article to represent the alternative views more accurately. Furthermore, we agree with the reviewer that SVs at common fragile sites are certainly understudied and this is the main aim of our commentary, to pique interest in this area to stimulate new research. Following review and suggestions from the Editor, we have altered this piece to be a Perspective to allow for more speculation. We hope you feel these changes enhance the article to a suitable standard.

Just as FS-CNA interaction is cell-type specific, it is likely that FS SV signatures are specific to certain tumour types, and this could prove to be a worthwhile area of research.

We agree, we have now mentioned this within the text (lines 160-165) to highlight that future work would benefit from a more in-depth analysis of specific tissue and tumour types.

Furthermore, because FS SV signatures are dominated by deletions (copy number loss) with well-described mechanistic models, the presence of copy number `_gains_` (tandem duplications) within FS is interesting, and possibly worthy of further investigation. However, while this paper provides a reasonable summary of existing work on FS and CNA in cancer genomics, and a call-to-arms for research into this area, it does not by itself add sufficient novelty to the field.

Thank you, as stated above the aim of this comment was not to posit an original hypothesis without supporting evidence, simply to highlight this interesting and understudied field in cancer genomics and place our recent work on the 3q26.31-32 locus within the wider emerging landscape. In writing this commentary, we aimed to summarise existing work which looks at fragile sites and SVs in cancer genomics and some of the suggested mechanisms for how SVs occur in these regions. To contextualise the significance of these sites we used *NAALADL2* as an example, which is surrounded by oncogenes. This also highlights that the function of the gene in which fragile sites occur may not be the only important factor. As with *NAALADL2* it may well be that genomic location is also important. We hope that through inclusion of the additional literature we have presented a more balanced view and that this will stimulate further research into these sites.

Two major claims were (re-)presented in this commentary, both of which were recently published by the same authors in the same journal (Simpson et al. 2020). Firstly, the authors reiterated that copy number gains in the FS gene *NAALADL2* were found in 16% of localised prostate cancer (PCa) and is associated with aggressive disease. The mention of this

claim is interesting in this commentary because it is not supported by the PCAWG (Li et al.) paper, which the authors heavily cited. In the PCAWG paper, CNA for NAALADL2 is similar to other FS genes in that SVs within FS are dominated by copy number losses rather than gains.

We agree, in their pan-cancer analysis Li *et al.*, showed deletions overall and within this region are more frequent. Future work will be needed to determine the clinical associations of SVs in this region in other tumour types. We have made it clear that in overall loss of NAALADL2 was more frequent than gains (lines 120-121) and discussed the need for tissue-specific investigations into these sites (lines 160-165)

Furthermore, no FS SV signature was observed in PCa in the PCAWG study (Sup Fig. 8).

We also agree that prostate cancer is not particularly enriched for SVs in fragile sites. It is entirely possible that not all fragile sites are important in all cancers. We have now highlighted in-text (line 164-165) that overall the FS signature showed little enrichment in prostate.

The second major claim is that copy number gains in the NAALADL2 locus co-occur with nearby oncogenes. Given linkage disequilibrium and the fact that genomic breaks at fragile sites could theoretically occur anywhere within a FS, it is unsurprising that there is co-occurrence of copy number gains of nearby genomic region. However, CNA does not always translate proportionately, or at all, to transcriptional changes. Thus, it is important to demonstrate whether there is *also* co-occurrence of increased gene/protein expression of nearby oncogenes.

Within our previous paper we demonstrated that of the 465 proximal genes in which gains frequently co-occurred 47.5% of genes displayed a significant increase at a transcriptional level compared to just 2.36% which were downregulated. This is in keeping with a gene-dosage effect. We have clarified this in-text on lines 124-128.

Fundamentally, the authors are hypothesising that replication stress contributes to the formation of FS, particularly at NAALADL2, thereby increasing the copy number of nearby oncogenes, thus conferring oncogenesis. To demonstrate the validity of this hypothesis, the authors will need to (1) demonstrate replication stress contributes to FS, not the other way around

We acknowledge that it could be entirely the other way around and had perhaps overrepresented this perspective. We have now softened the language implying this and made an argument for reverse causality on lines 141-142. We do not believe this view is at odds with those of experts within the field as inducers of replication stress have been shown to cause CNVs at common FS within the genome (discussed further below).

, and (2) that copy number gain of common FS (e.g. NAALADL2) results not only in increased expression of NAALADL2 but also nearby oncogenes.

In our study we found a statistically significant increase in NAALADL2 expression as well as increased expression of 47.5% of the surrounding co-amplified genes consistent with this hypothesis.

I am concerned that the proposed hypothesis that “replication stress may contribute to the formation of clinically significant FS” is overstated. Association is not causation. The data that’s presented only support that copy number gain in NAALADL2 is *correlated* with increased protein levels that have been implicated in DNA replication stress. It does not imply that replication stress induces copy number gain of NAALADL2. From existing theories of other FS, it is equally plausible that CN gain of NAALADL2 may be the cause of replication stress.

We agree that CN gain could certainly be a potential cause of replication stress rather than the other way around and we have incorporated this alternative perspective (line 141-142). We do argue however, that replication stress contributing to CNV formation at FS is commonly stated as such in the literature. In one of the papers suggested by the reviewer by Thomas Glover and Thomas Wilson, the authors title a section of the review “Replication stress induces CNVs”. We therefore did not feel this hypothesis conflicts with current opinion. In regards to 3q26.31 being clinically significant, we have softened our language suggesting that it may be ‘potentially’ clinically significant (line 139).

Similarly, the suggestion that gains at common FS can provide a fitness advantage, thereby cancer clonal selection, is in significant contrast to the alternate hypothesis that *loss* of common FS genes results in genomic instability causing CNA of cancer driver genes, which in turn confer selective advantage (e.g. see Saldivar et al. Plos Genetics, 2012; 8(11): e1003077). While it is important to challenge dogmas, it is important for the authors to recognise/acknowledge existing theory and to demonstrate whether it is complementary or mutually exclusive of the hypothesis they are proposing.

Respectfully, the authors disagree that our assertion that gains in specific regions may provide a fitness advantage contrasts the work of Saldivar *et al.* In fact, this view is complementary. Rather than loss or gains determining whether advantage or disadvantage is conferred, the gene(s) in which the SVs occur will confer a different selection advantage or disadvantage depending on their function.

Saldivar *et al.*, showed that loss of tumour suppressor *FHIT* increased genomic instability writing: “Our findings strongly support the view that loss of *FHIT* provides a selective advantage”. This finding is exactly in-line with the function of *FHIT*. A gain in this type of gene would likely reduce fitness and would not be selected for. We believe in the case of 3q26.31-32 as this region is rich in genes which, when overexpressed, act to enable proliferation this would confer advantage and loss would confer disadvantage. It is our belief that this view is in line with the current dogma.

Findings from the authors previous studies are a good starting point but further experiments / investigations are needed to support the current hypothesis.

If the authors are able to provide the suggested proofs, I would encourage a full article rather than a commentary.

Thank you for your comments and time, following discussion with the Editor we have expanded this to a Perspective piece to explore this hypothesis but also highlight opposing views and other possibilities. We believe this will still act to highlight this important area of research and inform the readers of Communications Biology. We hope you agree.

Specific points:

- The authors stated that “The FS signature showed little co-occurrence with alterations in DNA repair genes...”. This statement is not entirely correct as TP53 was shown to be mildly associated with 10 of the 12 most distinctive SV signatures in the PCAWG paper.

Thank you, we have now softened this language and changed “little co-occurrence” with “only mild co-occurrence”.

Furthermore, it is known (e.g. Schepeler et al. *Oncogene*, 2013; 32: 3577-3586) that TP53 mutations can provide the necessary replicative stress and selective pressure for the outgrowth of clones with defective DNA damage-response.

We agree Schepler *et al.*, did show some co-occurrence in bladder cancer however, as far as we can determine TP53 mutations status was inferred by immunohistochemistry where absent or very high staining both were considered to reflect a mutation while, intermediate staining was considered to reflect WT. This methodology is somewhat controversial as this may not necessarily accurately reflect TP53 mutation status. TP53 expression is subject to a number regulatory mechanisms at both the transcriptional and translational level. For this reason we did not represent strongly the evidence which suggests FS SVs are caused by DNA repair defects.

- Table 1 is supposed to present commonly affected FS highlighted by the PCAWG (Li et al.) paper. However, the original paper reported 18 major fragile sites (in their supplementary information), while 21 are reported in this Table 1. Presumably that’s because Table 1 was transcribed from Li et al’s Extended Data Fig. 9B, but nonetheless, clarification on the discrepancy would be helpful.

Yes, the reviewer is correct. We have now clarified where this data was retrieved from in the table legend.

- The authors indicated that “NAALADL2 was identified as the fourth most altered site in a pan-cancer analysis by Li et al., (2020)”. However, Extended Data Fig 9 of Li’s paper suggest NAALADL2 is the FS gene with the fifth most number of PCAWG samples (not fourth) and 6th to 8th most number of SVs (not fourth). Clarification on where the 4th ranking was derived would be helpful.

This was an error. We have now clarified this to fifth. Thank you.

- The authors noted that “Given the large size of NAALADL2, an increase in transcription prior to replication through SNPs and/or altered transcription factor binding could theoretically increase replication stress and the chance of breakage within the gene due to the FS within it.” However, wouldn’t “gene breakage” imply a loss of function or copy number loss?

As discussed in the article, while gene breakage may intuitively suggest a loss, the breakage–fusion–bridge (BFB) cycle model proposes that double strand breaks between the DNA is bridged, joining the Watson and Crick strands and that over progressive cycles of breakage and fusion, create a series of tandem inverted gene duplications which explain gains in these regions. Alternatively the microhomology-mediated break-induced replication model proposes that upon breakage, the 3’ end of the collapsed fork anneals to a single stranded

template with sufficient microhomology, which if annealing behind the replication fork creates a duplication. We have emphasised these points in-text.

Reviewer #2 (Remarks to the Author):

Simpson et al. present a nicely written summary of the specific class of copy number alterations seen in cancer that arise at chromosome fragile sites (FSs). It is timely, as these genes continue to gain increasing attention and yet are difficult to fully understand regarding the impacts of alterations. These loci are hyper mutagenic and so mutations may or may not correlate with oncogenic function. In this context, the perspective offers a nice balance between an initial general consideration and a latter portion that dives more deeply into what I agree is a less well considered gene in this context, NAALADL2, but which is clear a site of hyper-mutation. Overall, the review is clear, interesting and accurate and I endorse it.

Thank you for your kind words and endorsement of our article. We agree this is an interesting area of research that still receives relatively little attention and have aimed to highlight this topic and use the example of NAALADL2 as a gene which relatively understudied yet may be of importance. Following review and suggestions by the editor, we have expanded this commentary into a perspective piece. This can allow for a slightly more comprehensive literature review but also allow us to discuss alternative perspectives on some of the hypotheses given in this paper. We have also done our best to satisfy the constructive feedback you have provided and believe the article is of a higher quality as a result.

1) Abstract: the phrase “may span hundreds of genes” is confusing as it is hard to reconcile a focal process (at a single gene such as NAALADL2) with an event that would cover large portions of a chromosome. For the claim to be compelling, I would expect it to be possible to know that such very large events were mechanistically or functionally linked to the one FS gene of interest, and I am not sure that is possible. Simply changing to “many genes” may ameliorate a potentially misleading phrase.

Thank you, we have now altered this in-text.

2) Table 1 and associated text. It is disappointing that the presentation uses the GRCh37/hg19 reference genome, instead of hg38. Transcript annotations associated with hg38 are significantly more accurate at many large genes, in fact leading us to understand that they are even larger than as tabulated (one example is LSAMP). hg38 has been the standard reference genome from some years now.

Apologies, this was actually an in-text error, these chromosomal co-ordinates and gene lengths were in fact, produced using the hg38 build. This has now been updated to GRCh38.p13 in text and within the legend. Thank you for highlighting this.

3) Page 2, line 52. I do not agree that the TrDoFF model referenced by the authors suggests that “genomic instability may arise from cellular stress induced by HIGH levels of transcription”. In fact, Wilson et al. (ref 4) showed that the level of transcription did not correlate with instability in vitro, a concept extended by Blin et al (not referenced, PMID: 30598553) who showed that high levels of transcription in fact reduced instability. The thought could be fixed by simply removing “high levels of”.

Thank you for highlighting this. We have changed the wording as suggested and expanded the discussion of this point (lines 53-55). We have added the reference suggested to this section.

4) Page 2, line 62. To be sure, I agree that the mechanisms of duplication formation are enigmatic, but I find the reference to template switching to the opposite side of the gene to be somewhat at odds with the most nuanced understanding of the FoSTeS and MMBIR mechanisms. Newer papers from Lupski and colleagues that proposed these mechanisms suggest that duplications arise when template switching occurs to a location BEHIND the stalled fork of interest. Since the forks usually stall within the genes, this predicts that most FS-associated duplications will occur on the flanks of those genes, which was observed in vitro by Wilson et al. For a switch behind a fork to lead to duplication of an entire FS gene, the fork would have had to have already passed through the entire gene, at which point replication would likely have already been resolved. Thus, duplications that include the entirety of a FS gene likely occurred by a mechanism other than instability of the FS gene itself (such as some of those listed). I'm not sure this requires too much text alteration, but perhaps the sense of how and where template switching is likely to occur at FS genes would help with clarity.

We have discussed this interesting point more in-text as a perspective article has a larger word limit than a Commentary. We have removed sections which suggest increased transcription would increase the likelihood of breakage.

5) page 3 , line 92. "suggesting possible synergy". The fact that NAALADL2 genes are in disequilibrium with other genes associated with cancer might also support a hypothesis that one or the other are false positive associations, e.g. that the NAALADL2 alterations are not the drivers of an observed cancer association?

We agree that NAALADL2 SNPs may simply be a bystander or indeed false positive, we have now added this in-text (lines 109-110).

6) There are two points from the literature that are not referenced that I think are highly relevant to the discussion. I would recommend working them in if possible:

a) Wei et al (PMID: 26871630) used a DSB-driven assay to show that large genes are frequent sites of genomic alterations such as translocations and others. It stands as an excellent documentation of their increased instability in primary cells.

b) Macheret et al (PMID: 32561860) and others recently showed that large genes at FSs are uniquely prone to completing replication in M phase, known as Mitotic DNA Synthesis (MiDAS). I don't consider any discussion of FS instability any more without considering that highly exotic replication pattern.

Thank you for these great suggestions, both of these references have now been incorporated.

Although overall the paper is clearly written, I point out a few places where I felt the grammatical usage could be improved:

a) page 1, line 31, “replication stress acting as genomic ‘fault lines’”. Sounds like replication stress is acting as a fault line (not the FS). Probably just put a comma after “stress”.

Thank you. This is now changed in text.

b) page 3, line 86. “SNPs within the NAALADL2 locus associated with prostate cancer”. Given the sentence structure, I might suggest adding “that are associated” to make it clear that it is the SNPs that are associated, not the gene.

Thank you. This is now changed in text.

c) page 3, line 82, “However, more frequently, supplementary data exists”. I suggest deleting “more frequently, supplementary”. It is not clear what is meant to “more frequent” or “supplementary” so those words seem superfluous. Also, “data” is a plural word, so it should be “exist”, not “exists”.

Here we were trying to imply that few manuscripts will mention NAALADL2 within the main body of the paper or even highlight this gene, even if it is highly significant or in some cases upregulated. This may be because its function and significance is simply not known. That being said, it is often within the supplementary data, buried within tables. However, your suggestion is useful and for simplicity, we have made some changes.

d) page 3 , line 90. “BCR” is never defined, I don’t know what it is.

Thank you, we have now defined this in-text.

e) page 3 line 105. “NAALADL2 gains associated with”. Add “were” before “gains”. In general, you will see that I think the liberal use of the word “associated” was sometimes confusing without more words to guide the grammar.

This has now been changed as instructed, thank you for the suggestion.

REVIEWERS' COMMENTS:

Reviewer #1 (Remarks to the Author):

The Authors' positive Response to the Review is welcoming.

The original Commentary certainly gave (at least me) the impression that the Authors were attempting to propose a theory that is mutually exclusive of existing hypotheses. Clarifications in the Response have been helpful and I agree that a Perspective is more suitable for this manuscript, as are the inclusion of a more diverse range of literatures. Clarifications of the Authors' previous published work have also helped with contextualisation.

The Response regarding immunochemistry staining for TP53 mutation is interesting as I had not appreciated (perhaps I should) the nuances of this method.

Overall, the Authors have adequately addressed my Comments and I would support it be published.

Reviewer #2 (Remarks to the Author):

First, my apologies for being a bit late with this review – the end of our academic term was very challenging this year especially.

Simpson et al. present a somewhat longer version of their paper considering the general association between fragile site CNVs and cancer and a more specific consideration of NAALADL2. I was previously positive about the paper but had a number of comments. I think the authors were strongly responsive to those, with the benefit of having some additional words, and that the paper is improved. I continue to endorse it.

I see that Reviewer 1 was more critical and appears to have a more detailed working knowledge of the genomic findings in cancer, e.g. in PCAWG, than I do. I was troubled that he/she seemed to dispute the accuracy of some of the claims made. I note that the authors have attempted to use additional words to provide a more substantive and accurate description and trust that they are properly reflecting the literature on those points.

I continue to see this paper as a "perspective", in force if not in name. I don't think it is able to draw hard conclusions and I am less concerned that it establishes novel findings. It provides a nice introduction to the main concepts in the field and raises interesting questions about NAALADL2, which I agree is not as well studied. The weaknesses continue to reside in the degree of specificity with which cause and effect associations can be made, some of which were pointed out by each reviewer. For me, perhaps the most limiting remain the very broad regional effect being connected to a single FS gene and the suggestion that FS are a primary driver of gains when it is known that deletions are more typical. However, in the end, I think these limitations are tolerable in a paper that uses balanced language to suggest an interesting area for future study more than a claim of hard results.